# Protective Effects of Betanin in Acute and Subacute Periods in Experimental Colitis Induced by Trinitrobenzene Sulfonic Acid

**DOI:** 10.3390/nu18010086

**Published:** 2025-12-26

**Authors:** Ismail Taskiran, Adem Keskin, Adil Coskun, Ferhat Sirinyildiz, Ibrahim Meteoglu

**Affiliations:** 1Department of Gastroenterology, Faculty of Medicine, Aydin Adnan Menderes University, Aydin 09100, Turkey; adilcoskun@gmail.com; 2Department of Medical Biochemistry, Faculty of Medicine, Aydin Adnan Menderes University, Aydin 09100, Turkey; adem.keskin@adu.edu.tr; 3Department of Physiology, Faculty of Medicine, Aydin Adnan Menderes University, Aydin 09100, Turkey; ferhat.sirinyildiz@adu.edu.tr; 4Department of Pathology, Faculty of Medicine, Aydin Adnan Menderes University, Aydin 09100, Turkey; imeteoglu@hotmail.com

**Keywords:** ulcerative colitis, trinitrobenzene sulfonic acid, betanin, antioxidant, anti-inflammatory, acute, subacute, inflammation, oxidative stress, histopathology

## Abstract

**Background/Objectives**: This study aims to investigate the effects of betanin, which possesses antioxidant and anti-inflammatory properties, against colitis, a disease in which inflammation and oxidative stress play a role in its pathophysiology, during the acute and subacute period. **Methods**: Thirty-two rats were included in the study and divided into four groups: control, colitis, 3-day betanin supplementation + colitis (bet3+colitis), and 15-day betanin supplementation + colitis (bet15+colitis). Experimental colitis was induced with trinitrobenzene sulfonic acid. **Results**: In the colitis group, malondialdehyde, myeloperoxidase, superoxide dismutase (SOD) inhibition rate, tumor necrosis factor-alpha (TNF-α), interleukin-1 beta (IL-1β), interleukin-6, mucosal damage, and cell infiltration scores were higher than in the control group, while catalase and glutathione peroxidase (GPx) levels were lower. In the betanin supplementation groups, malondialdehyde, myeloperoxidase, TNF-α, IL-1β, mucosal damage, and cell infiltration scores were lower than in the colitis group, while GPx levels were higher. In addition, the SOD inhibition rate and interleukin-6 levels were lower in the bet15+colitis group than in the colitis group. TNF-α, IL-1β, interleukin-6, and GPx levels in the betanin groups were similar to the control group. **Conclusions**: Betanin supplementation demonstrated a significant anti-inflammatory effect by reducing inflammatory parameters and histopathological scores in both periods. Additionally, it exhibited a glutathione-related antioxidant effect by increasing GPx levels in both periods. However, although SOD inhibition rates decreased in the subacute period, no significant change in catalase levels was observed in either period. This indicates that it did not provide complete protection in terms of antioxidant effects in either period.

## 1. Introduction

Ulcerative colitis, a lifelong inflammatory disease affecting the rectum and colon to varying degrees, typically presents with bloody diarrhea and is diagnosed based on a combination of clinical, biological, endoscopic, and histological findings. Ulcerative colitis, whose pathophysiology is strongly influenced by intestinal epithelial barrier defects, microbiota, and dysregulated immune responses, has an estimated 5 million cases worldwide, and its prevalence is increasing [1]. Western dietary habits, particularly prevalent in newly industrialized countries, along with certain medications and lifestyle factors that can influence the host microbiota or the immune response to antigens, play a significant role in the development of this disease. While there is strong evidence that genetic and host factors contribute to disease susceptibility, many mechanisms related to the pathophysiology of ulcerative colitis have not yet been fully elucidated [2]. Because the etiology and underlying mechanisms of the disease are not yet fully understood, there is no definitive treatment or “gold standard” diagnostic method for this disease. Current treatments generally aim to relieve symptoms and control inflammation, but they cannot completely eliminate the disease. Therefore, patients with ulcerative colitis often face a reduced quality of life and an increased risk of developing colorectal cancer [3].

Oxidative stress, a key determinant of ulcerative colitis pathogenesis, plays a critical role in both the initiation and maintenance of inflammation by triggering an inflammatory response, sustaining immune cell activation, and causing structural damage to cellular components [4]. Oxidative stress-causing reactive oxygen species, such as superoxide anion free radicals, hydrogen peroxide, and hydroxyl radicals, are excessively produced during the inflammatory process, leading to protein dysfunction, DNA damage, and lipid peroxidation. These reactive oxygen species activate cellular signaling pathways, creating a vicious cycle of oxidative damage [5]. This vicious cycle is not limited to local tissue damage but can also extend its effects systemically, increasing the risk of carcinogenesis. Accordingly, targeting oxidative stress in the treatment of ulcerative colitis is considered a promising strategy to prevent inflammation-induced DNA damage and damage associated with malignant transformation [6]. Therefore, numerous molecules and natural products with antioxidant properties have become the focus of recent studies investigating their potential therapeutic effects on ulcerative colitis [4,7,8,9,10,11,12].

Betanin, one of the main pigments of red beet, stands out as a natural component with various protective and therapeutic effects thanks to its biologically active structure [13]. Furthermore, betanin is a non-toxic, natural colorant approved by the Food and Drug Administration (FDA) and the European Union and is authorized for use under quantum satis with code E162 [14]. Additionally, betanin acts as a multi-pathway modulator of oxidative stress and inflammation, offering protective effects against disease pathogenesis [15]. Therefore, betanin’s antioxidant and anti-inflammatory properties have been investigated in a wide range of studies, ranging from certain types of cancer to diabetes, neurological, and cardiovascular diseases, with promising results [16,17,18,19,20,21,22].

Inflammation and oxidative stress are two key features of ulcerative colitis pathophysiology. On the other hand, betanin is a natural compound with antioxidant and anti-inflammatory properties. This study aimed to evaluate the antioxidant and anti-inflammatory effects of short-term (acute, 3 days) and long-term (subacute, 15 days) betanin supplementation on oxidative stress, antioxidant capacity, and inflammation parameters in an experimental colitis model induced by trinitrobenzene sulfonic acid. Histopathological changes in colonic tissue, including mucosal damage and cellular infiltration, were also included in the evaluation criteria.

## 2. Materials and Methods

The necessary ethics committee approval for the study was received from the Aydın Adnan Menderes University Experimental Animal Ethics Committee on 31 May 2021 (Approval number 64583101/2021/072). All experimental procedures were conducted in strict accordance with relevant guidelines and regulations, including the ARRIVE 2.0 guideline and the 2010/63/EU Directive on the Protection of Animals for Scientific Purposes [23].

The study was conducted with 32 female Wistar Albino rats, aged 8–10 months and weighing 200–250 g. Throughout the study, rats were kept in an environment with a controlled ambient temperature of 22 ± 2 °C, 65% average humidity. In addition, rats were housed on a 12 h light/12 h dark cycle, with lights on at 8 a.m. and off at 8 p.m. Additionally, all rats were provided with standard laboratory feed and tap water ad libitum throughout the experiment. Feeding and experimental procedures were performed according to these standard conditions. The rats were randomly assigned to 4 groups:Control group: No supplementation and no experimental colitis induced (*n*: 8);Colitis group: No supplementation and experimental colitis induced (*n*: 8);Bet3+colitis group: 50 mg/kg betanin supplementation for 3 days followed by experimental colitis induction (*n*: 8);Bet15+colitis group: 50 mg/kg betanin supplementation for 15 days followed by experimental colitis induction (*n*: 8).

### 2.1. Acute and Subacute Period Administration of Betanin Supplementation

Betanin supplement (red beetroot extract diluted with dextrin) (product code 901266, Sigma-Aldrich Chimie, Saint-Quentin-Fallavier, France) was supplied by Sigma-Aldrich [24]. The betanin supplement was diluted to the appropriate concentration with sterile water before use and administered via intragastric gavage at a dose of 50 mg/kg to the bet3+colitis and bet15+colitis groups. The gavage volume was adjusted to 1 mL/100 g body weight to ensure accurate dosing across animals. The administered dose was determined based on previous studies examining the antioxidant and anti-inflammatory properties of betanin [25,26,27,28]. The duration of betanin supplementation was determined by its potential effect. In previous studies, administration for 7 days or less was classified as acute, and administration between 7 and 28 days was classified as subacute [29,30,31,32]. Accordingly, betanin supplementation was administered for 3 days to the bet3+colitis group to assess its acute effect, and for 15 days to the bet15+colitis group to assess its subacute effect.

### 2.2. Experimental Colitis Induction

Before experimental colitis induction, all rats were fasted for 24 h, and the colon was emptied by triggering the defecation reflex. Experimental colitis induction was performed after supplementation in the Bet3+colitis and Bet15+colitis groups; the colitis group underwent the same induction procedure. The experimental colitis induction procedure was performed under 75 mg/kg ketamine (Ketalar^®^; Parke Davis, Eczacıbaşı, Istanbul, Turkey) and 8 mg/kg xylazine (Rompun^®^; Bayer AG, Leverkusen, Germany) anesthesia, a polyethylene cannula was inserted 8 cm through the anus, and 25 mg of trinitrobenzene sulfonic acid dissolved in 0.8 mL of 37% ethanol was administered intrarectally to induce colitis. Trinitrobenzene sulfonic acid selection and induction protocol were based on previous studies. The control group also underwent the experimental colitis induction procedure but received physiological saline instead of trinitrobenzene sulfonic acid. Rats were sacrificed with tissue samples taken 72 h after experimental colitis induction.

Euthanasia was performed by cervical dislocation under 80 mg/kg ketamine (Ketalar^®^; Parke Davis, Eczacıbaşı, Istanbul, Turkey) and 10 mg/kg xylazine (Rompun^®^; Bayer AG, Leverkusen, Germany) anesthesia. A 10-cm segment of the colon was excised from each rat. The collected colons were opened longitudinally, and the tissues were gently rinsed with physiological saline. The cleaned colon tissue was divided lengthwise into two equal parts, and this procedure was performed based on randomly selected sections throughout the tissue. While one half was homogenized for biochemical analysis, the other half was fixed appropriately for histopathological examination. Thus, comparable and representative tissues taken from the same colon were used for both analyses.

### 2.3. Biochemical Processes

The colon tissue samples were washed with 0.9% NaCl and homogenized in 50 mM phosphate buffer, pH 7.0, at 4 °C. Homogenates were centrifuged at 15,000 rpm for 15 min at 4 °C and the supernatants were separated for biochemical analysis.

Malondialdehyde (Catalog No: K739-100, Biovision Incorporated, Milpitas, CA, USA), interleukin-1 beta (IL-1β) (Catalog No: K4794-100, Biovision Incorporated), interleukin-6 (IL-6) (Catalog No: K4143-100, Biovision Incorporated), tumor necrosis factor-alpha (TNF-α) (Catalog No: 3052R-100, Biovision Incorporated), myeloperoxidase (Catalog No: K744-100, Biovision Incorporated), glutathione peroxidase (GPx) (Catalog No: K762-100, Biovision Incorporated), catalase (Catalog No: K773-100, Biovision Incorporated) and superoxide dismutase (SOD) (inhibition rate) (Catalog No: K335-100, Biovision Incorporated) enzyme activities were measured by colorimetric/fluorometric methods according to the manufacturer’s protocol.

### 2.4. Histopathological Analysis

Colon tissues were treated with 10% neutral formaldehyde. The fixation temperature was 4 °C and the duration was 24 h. The tissues were dehydrated with ethanol and cleared in xylene. The tissues were embedded in paraffin blocks, and 5-µm-thick random sections were cut using a microtome (Leica model RM 2135, Leica, Wetzlar, Germany). The sections were stained with hematoxylin-eosin (H&E). A light microscope was used to examine the stained sections, and screenshots were taken with a digital camera (Olympus DP20, Tokyo, Japan) connected to a microscope (Olympus BX51, Tokyo, Japan). Histopathological changes in the parameters were evaluated and scored accordingly.

An adapted semi-quantitative panel was used to assess mucosal damage (0–3) and cellular infiltration (0–3) in colon tissue. Cellular infiltration was scored as follows: a score of 0 indicates preservation of basal cellularity in the lamina propria; A score of 1 indicates lamina propria infiltration limited to a mild leukocyte increase; a score of 2 indicates moderate infiltration with a multifocal distribution including pericryptal areas; and a score of 3 indicates diffuse and intense inflammatory infiltration with crypt compression, cryptitis/crypt abscesses, and marked mucosal architectural disruption. In the assessment of mucosal damage, a score of 0 indicates normal epithelial integrity; a score of 1 indicates mild degeneration or focal loss of the surface epithelium; a score of 2 indicates moderate damage including partial crypt loss, epithelial erosions, and occasional architectural disruption; and a score of 3 indicates widespread epithelial loss, ulceration, marked crypt loss, and severe mucosal destruction extending into the submucosa.

As part of the microscopic histopathological evaluation, the pathologist prepared eight different sections from each tissue sample and took ten microtome samples from each section. After staining these samples, a random sample was selected from each section group for microscopic examination. Since the 10 cm colon segment taken as part of the experimental protocol included both the proximal and distal colon regions, the randomly selected sections contained samples from both anatomical regions. The examinations showed that trinitrobenzene sulfonic acid administration produced histopathological findings associated with ulcerative colitis in a similar manner in both the proximal and distal colon regions. All evaluations were performed using the same longitudinal section analysis method.

### 2.5. Statistical Analysis

SPSS for Windows 22.0 program was used for statistical analysis. Continuous variables obtained in the study were expressed as mean ± standard deviation (X ± SD), and a *p* value below 0.05 indicates that the difference between the values is statistically significant. The normality of the data distribution was assessed using the Kolmogorov–Smirnov normality test with Lilliefors correction. Normally distributed data were compared using One-Way ANOVA. The selection of the Post Hoc Test, which allows determining the significance levels of differences between groups, was decided by testing variance homogeneity with the Levene test. Considering that the sample numbers in the groups were equal, the Tukey HSD test was used for data that provided variance homogeneity with the Levene test, and the Dunnett T3 test, a Post Hoc Test, was used for data that did not.

## 3. Results

### 3.1. Effects of Trinitrobenzene Sulfonic Acid-Induced Experimental Colitis on Oxidant, Antioxidant and Inflammatory Parameters

In the colitis group, oxidant parameters such as malondialdehyde and myeloperoxidase, antioxidant parameters such as SOD inhibition rate and inflammation markers such as TNF-α, IL-1β and IL-6 were found to be higher than in the control group, while antioxidant enzyme activity values such as catalase and glutathione peroxidase (GPx) were found to be lower (*p* < 0.001), (Table 1).

### 3.2. Effects of Acute Period Betanin Supplementation on Oxidant, Antioxidant, and Inflammatory Parameters

In the bet3+colitis group, oxidant parameters such as malondialdehyde and myeloperoxidase and inflammation markers such as TNF-α and IL-1β were found to be lower than in the colitis group, while antioxidant enzyme activity values such as GPx were found to be higher (*p* < 0.001, *p* < 0.001, *p* < 0.001, *p* = 0.004, *p* < 0.001, respectively) (Table 1). No significant difference was found between the two groups in terms of SOD inhibition rate, catalase and IL-6 levels (*p* = 0.388, *p* = 0.214, *p* = 0.067, respectively) (Table 1).

In the bet3+colitis group, oxidant parameters such as malondialdehyde and myeloperoxidase and antioxidant parameters such as SOD inhibition rate were found to be higher than in the control group, while antioxidant enzyme activity values such as catalase were found to be lower (*p* < 0.001) (Table 1). No significant difference was found between the two groups in terms of TNF-α, IL-1β, IL-6 and GPx levels (*p* = 0.999, *p* = 0.064, *p* = 0.051, *p* = 0.111, respectively) (Table 1).

### 3.3. Effects of Subacute Period Betanin Supplementation on Oxidant, Antioxidant, and Inflammatory Parameters

In the bet15+colitis group, oxidant parameters such as malondialdehyde and myeloperoxidase, antioxidant parameters such as SOD inhibition rate and inflammation markers such as TNF-α, IL-1β and IL-6 were found to be lower than in the colitis group, while antioxidant enzyme activity values such as GPx were found to be higher (*p* < 0.001, *p* < 0.001, *p* = 0.001, *p* < 0.001, *p* < 0.001, *p* = 0.001, *p* < 0.001, respectively) (Table 1). No significant difference was found between the two groups in terms of catalase levels (*p* = 0.568) (Table 1).

In the bet15+colitis group, oxidant parameters such as malondialdehyde and antioxidant parameters such as SOD inhibition rate were found to be higher than in the control group, while antioxidant enzyme activity values such as catalase were found to be lower (*p* < 0.001, *p* = 0.001, *p* < 0.001, respectively) (Table 1). No significant difference was found between the two groups in terms of myeloperoxidase, TNF-α, IL-1β, IL-6 and GPx levels (*p* = 0.052, *p* = 0.543, *p* = 0.294, *p* = 0.662, *p* = 0.098, respectively) (Table 1).

### 3.4. Effects of Subacute Period Betanin Supplementation on Oxidant, Antioxidant, and Inflammatory Parameters Compared to Acute Period Betanin Supplementation

No significant difference was found between the bet15+colitis and bet3+colitis groups in terms of malondialdehyde, myeloperoxidase, SOD inhibition rate, catalase, GPx, TNF-α, IL-1β and IL-6 levels (*p* = 0.568, *p* = 0.205, *p* = 0.097, *p* = 0.900, *p* = 1.000, *p* = 0.456, *p* = 0.848, *p* = 0.412) (Table 1).

### 3.5. Effects of Acute and Subacute Periods Betanin Supplementation on Histological Findings in Trinitrobenzene Sulfonic Acid-Induced Experimental Colitis

No mucosal damage was detected in any rat in the control group (Figure 1), while all rats in the colitis group showed severe mucosal destruction (Figure 2). Similarly, in terms of cellular infiltration, all rats in the colitis group exhibited widespread and intense inflammatory infiltration (Figure 2), while only two rats in the control group exhibited limited lamina propria infiltration with a mild leukocyte increase (Figure 1). Mucosal damage and cellular infiltration scores for all groups are presented in Table 2.

In all experimental colitis induced groups, mucosal damage and cellular infiltration scores were found to be higher than those in the control group (*p* < 0.001) (Table 2). Mucosal damage and cellular infiltration scores of the Bet3+colitis group were found to be lower than those in the colitis group (*p* = 0.021) (Table 2) (Figure 3). Mucosal damage and cellular infiltration scores of the Bet15+colitis group were found to be lower than those in the colitis group (*p* < 0.001) (Table 2) (Figure 4). Cellular infiltration scores of the Bet15+colitis group were found to be lower than those in the Bet3+colitis group (*p* = 0.046) (Table 2) (Figure 4).

## 4. Discussion

Malondialdehyde and other reactive products formed during lipid peroxidation, which play a key role in the pathological mechanisms associated with colitis, worsen the pathological course of the disease through various mechanisms, including disrupting cell membrane integrity, triggering apoptosis/necrosis, and activating inflammatory signaling pathways. Furthermore, these products, which increase in intestinal epithelial cells during disease, can increase intestinal permeability by weakening barrier functions, allowing pathogenic bacteria and toxins to readily enter the systemic circulation [33]. More importantly, this barrier dysfunction creates a mechanical bridge between oxidative stress and mucosal inflammation: when permeability increases, luminal antigens further enhance neutrophil aggregation and cytokine production, creating a self-sustaining “oxidation-inflammation” cycle that accelerates tissue damage.

In a recent study using experimental colitis models induced in vivo and in vitro in mice with sodium dextran sulfate, it was reported that malondialdehyde and myeloperoxidase levels were significantly increased in colitis-affected mice compared to healthy controls, and that ferroptosis plays an indispensable role in the pathological process of colitis [34]. Similarly, another recent study conducted on rats induced with trinitrobenzene sulfonic acid reported that malondialdehyde and myeloperoxidase levels were markedly increased in colitic rats compared to healthy controls [11]. On the other hand, in a recent study in which 100 mg/kg betanin supplementation was administered intraperitoneally as a single dose, it was found that malondialdehyde and myeloperoxidase levels in betanin-supplemented rats were significantly decreased compared to healthy rats [35]. Additionally, another recent study in which 100 mg/kg betanin was administered orally for 7 days found that malondialdehyde levels were reduced in rats receiving betanin supplements compared to healthy rats [36]. Overall, the literature suggests that betanin can counteract lipid peroxidation and neutrophil-derived oxidative load; however, the magnitude of the benefit may vary depending on the dose, route of administration, and duration; this is important when interpreting the differences between acute and subacute supplementation protocols.

In this study, malondialdehyde levels were higher in colitis-affected rats compared to healthy rats, but lower in betanin-supplemented rats compared to other colitis-affected rats. Additionally, myeloperoxidase levels were higher in acute betanin-supplemented and unsupplemented colitis-affected rats compared to healthy rats, but myeloperoxidase levels in betanin-supplemented rats were lower than in other colitis-affected rats. Additionally, myeloperoxidase levels in subacute betanin-supplemented rats and healthy rats were similar. These findings suggest that betanin supplementation can reduce oxidative stress and neutrophil infiltration and/or activation even in the acute period; however, longer-term supplementation is required for normalization of myeloperoxidase.

Experimental colitis increases oxidative stress by permanently suppressing endogenous SOD activity [37]. With the excessive increase in reactive oxygen species in colitis, SOD is rapidly depleted to meet this oxidative load, and enzyme activity decreases [38]. Endogenous SOD deficiency increases reactive oxygen species production and reduces antioxidant enzyme activity. It also increases the infiltration of proinflammatory immune cells. Antioxidant supplementation improves acute colitis by correcting SOD deficiency [39]. In this context, a recent study reported that a single oral dose of 110 mg/kg of betanin, an antioxidant substance, stimulated SOD activation and prevented experimentally induced oxidative stress [13]. Beyond direct radical scavenging, an additional possible explanation is that betanin indirectly protects endogenous antioxidant defense by reducing upstream oxidative load; this helps prevent the oxidative inactivation of enzymes such as SOD during ongoing inflammation.

In this study, SOD inhibition rates were higher in colitis-affected rats compared to healthy rats, but lower in subacute betanin-supplemented rats compared to unsupplemented rats. The increase in SOD inhibition rate reflects the decrease in SOD activity. In other words, the excessive oxidative stress occurring in colitis exceeds the normal superoxide scavenging capacity of SOD, leading to oxidative modification or inactivation of the enzyme. Consequently, as SOD activity decreases, the measured SOD inhibition rate increases in parallel. On the other hand, betanin supplementation administered during the subacute period partially corrects this situation and alleviates the decrease in SOD activity.

In a recent study, it was reported that catalase enzyme activity values decreased by 42% and GPx enzyme activity values decreased by 44% in experimental colitis induced by dextran sulfate sodium compared to healthy controls [40]. Similarly, in another recent study, it was reported that decreases in catalase and GPx enzyme activities were observed in rats with experimental colitis induced with acetic acid compared to healthy rats [41]. On the other hand, in a study examining the antioxidant effect of betanin against experimentally induced nephrotoxicity, it was reported that oral supplementation at a dose of 20 mg/kg for 90 days (chronic period) provided improvement in catalase and GPx enzyme activity values [42]. Similarly, in a study examining the antioxidant effect of betanin against experimentally induced diabetes, oral supplementation was administered at doses of 10, 20 and 40 mg/kg for 28 days (subchronic period) and increases in catalase and GPx enzyme activity values were reported compared to diabetic controls [19]. These comparisons suggest that catalase sensitivity may be more sensitive to the disease model and duration of antioxidant exposure compared to GPx, which may recover earlier as part of the glutathione-dependent detoxification pathway.

In this study, catalase levels were lower in colitis-affected rats compared to healthy rats. Additionally, catalase levels in betanin-supplemented rats and unsupplemented colitis-affected rats were similar. GPx levels were lower in unsupplemented colitis-affected rats compared to healthy rats. In contrast, GPx levels in betanin-supplemented rats and healthy rats were similar. In addition, GPx levels were higher in betanin-supplemented rats compared to other colitis-affected rats. These findings indicate that betanin supplementation is insufficient to improve catalase enzyme activity in both the acute and subacute periods. Based on data reported in the literature, it is thought that longer-term administration of betanin supplementation may be necessary to achieve a significant improvement in catalase enzyme activity. In contrast, it has been determined that even the administration of betanin supplementation in the acute period is sufficient to improve GPx activity. This suggests that betanin primarily supports the glutathione-related detoxification system.

TNF-α, which plays a causal role in the pathogenesis of colitis, acts as a central regulator in initiating and sustaining excessive inflammatory responses in the intestinal mucosa. However, the excessive TNF produced at the onset of colitis inhibits bile acid detoxification in intestinal epithelial cells, leading to bile acid overload and consequently apoptosis. Elevated TNF-α levels in colitis also contribute to the disruption of epithelial barrier integrity, proinflammatory activation of macrophages and other immune cells, and progression of tissue damage [43,44]. Increased TNF-α levels disrupt the intestinal immune barrier while also promoting stem cell formation of Dclk1+ tuft cells, which serve as cancer cell lineages [45,46]. Furthermore, this cytokine elevation promotes the transformation of CD4^+^ T cells into pathogenic T cells that produce granulocyte-macrophage colony-stimulating factor in vivo, thus promoting the development of colitis and colorectal cancer [47]. Many recent studies have reported that betanin supplementation reduces the levels of inflammatory cytokines such as TNF-α, IL-1β and IL-6 in various conditions, from stomach ulcers to testicular toxicity, from acute kidney injury to alcoholic liver disease [48,49,50,51,52].

In this study, TNF-α, IL-1β and IL-6 levels were higher unsupplemented colitis-affected rats compared to healthy rats. In contrast, TNF-α, IL-1β and IL-6 levels in betanin-supplemented rats and healthy rats were similar. In addition, TNF-α and IL-1β levels were lower in betanin-supplemented rats compared to other colitis-affected rats. Moreover, IL-6 levels were lower in subacute betanin-supplemented rats compared to unsupplemented colitis-affected rats. These findings demonstrate that betanin supplementation exhibits anti-inflammatory effects even in the acute phase; however, longer-term supplementation is necessary for complete recovery in IL-6 levels.

In the colonic mucosa of ulcerative colitis patients, there is a marked infiltration of plasma cells, T and B lymphocytes, macrophages, mast cells, eosinophils, and neutrophil leukocytes in the superficial regions of the epithelium, crypts, and lamina propria compared to normal conditions. The number and size of lymphoid follicles in the lamina propria increase. With increased disease activity: Inflammatory cell infiltration within the mucosa increases, the density of the infiltrate rises, inflammatory changes in the crypts and the number of crypt abscesses increase, focal infiltration in the lamina propria decreases while diffuse infiltration increases, inflammatory cells spread from the surface to deeper layers of the lamina propria, and all cellular infiltrates, especially plasma cells, T lymphocytes, macrophages, and neutrophil leukocytes, become more prominent [53]. Furthermore, the histopathological diagnosis of colitis is based on the evaluation of structural damage in the intestinal mucosa, determination of the chronic inflammation pattern, and the characteristic cellular composition of the inflammatory infiltrate [54].

In this study, mucosal damage and cellular infiltration scores were higher in colitis-affected rats compared to healthy rats, but lower in betanin-supplemented rats compared to other colitis-affected rats. These findings indicate that betanin supplementation has a beneficial effect on histopathological scores in both acute and subacute periods; however, longer-term supplementation is thought to be necessary to achieve a completely healthy and normal histological appearance.

These histological improvements are consistent with the biochemical profile observed in the supplementation groups (reduced lipid peroxidation and lower inflammatory mediators) and support the common interpretation that betanin alleviates colitis severity by combining its antioxidant and anti-inflammatory effects. From our perspective, the most important clinical implication is that betanin supplementation appears to be more effective in normalizing inflammation-related markers (e.g., myeloperoxidase and IL-6) in the subacute phase, suggesting that sustained supplementation may be necessary to translate its antioxidant potential into robust mucosal protection.

This study has some limitations. First, the absence of a positive control group containing commonly used anti-inflammatory drugs or other known anti-colitis agents in the treatment of ulcerative colitis is a significant limitation. Second, the fact that the study was conducted only on female rats limits the generalizability of the findings to both sexes. Finally, the inability to directly measure molecular signaling mechanisms via NF-κB and Nrf2 pathways can be considered another limitation of the study. On the other hand, one of the strengths of this study is that it can serve as a reference for future studies where these limitations are overcome. In addition, being the first validation study of betanin efficacy in a colitis model, the simultaneous examination of acute and subacute periods, and the combined evaluation of oxidant-antioxidant, inflammatory, and histopathological parameters are other strengths of the study. 

## 5. Conclusions

Betanin supplementation has demonstrated anti-inflammatory effects by reducing inflammatory markers, antioxidant effects by lowering malondialdehyde levels and increasing GPx enzyme activity, and significant improvements in histopathological scores. Furthermore, a more pronounced decrease in SOD inhibition rate and myeloperoxidase levels in the subacute period indicates that betanin’s antioxidant effect is particularly enhanced during the subacute period. On the other hand, the absence of any effect on catalase levels in the acute and subacute periods, but the reporting of an effect in subchronic and chronic period studies, indicates that betanin’s effect on catalase is more pronounced with long-term application.

## Figures and Tables

**Figure 1 nutrients-18-00086-f001:**
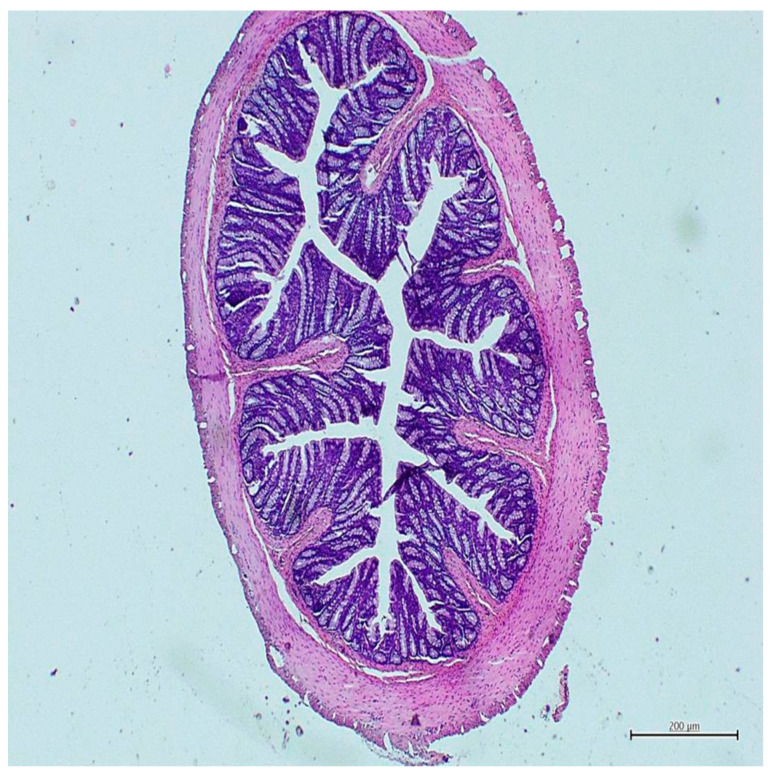
Histological image of the control group (H&E, ×40).

**Figure 2 nutrients-18-00086-f002:**
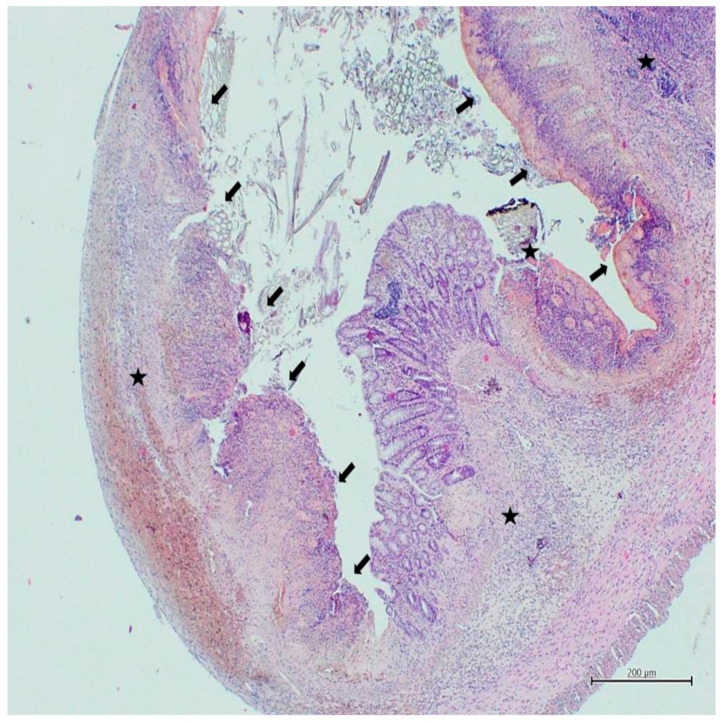
Histological image of the colitis group (H&E, ×40). Arrows indicate areas of mucosal damage with surface disruption and intraluminal debris, while asterisks indicate areas of dense inflammatory cell infiltration within the colon wall.

**Figure 3 nutrients-18-00086-f003:**
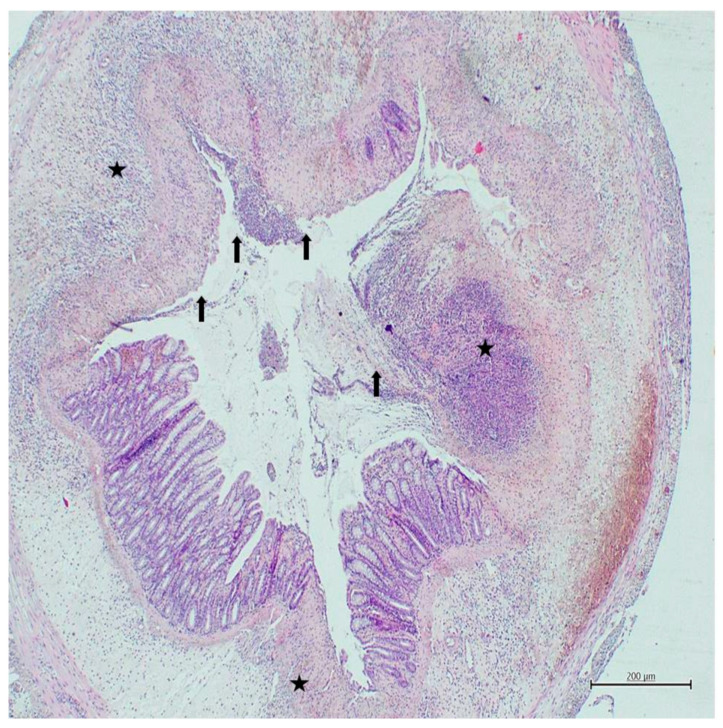
Histological image of the bet3+colitis group (H&E, ×40). Arrows indicate areas of mucosal damage with surface disruption and intraluminal debris, while asterisks indicate areas of dense inflammatory cell infiltration within the colon wall.

**Figure 4 nutrients-18-00086-f004:**
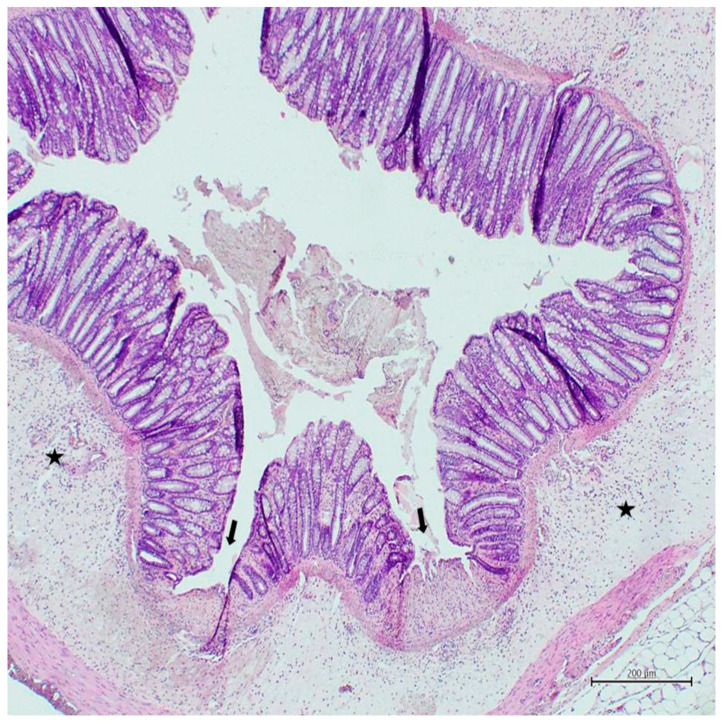
Histological image of the bet15+colitis group (H&E, ×40). Arrows indicate areas of mucosal damage with surface disruption and intraluminal debris, while asterisks indicate areas of dense inflammatory cell infiltration within the colon wall.

**Table 1 nutrients-18-00086-t001:** Oxidative stress, inflammation and antioxidant parameter levels of the groups.

Parameters	Control *n* = 8	Colitis *n* = 8	Bet3 + Colitis *n* = 8	Bet15 + Colitis *n* = 8
MDA (nmol/g)	77.25 ± 16.76 ^a^	227.25 ± 35.92 ^b^	154.63 ± 20.01 ^c^	137.5 ± 27.99 ^c^
MPO (U/g)	381.5 ± 85.62 ^a^	774 ± 79.98 ^b^	579.63 ± 66.71 ^c^	495 ± 99.57 ^a,c^
SOD (%)	18.75 ± 1.98 ^a^	53.88 ± 7.36 ^b^	47.13 ± 11.59 ^b,c^	37 ± 9.38 ^c^
Catalase (U/g)	180.13 ± 24.4 ^a^	102.13 ± 20.26 ^b^	121.63 ± 19.15 ^b^	114.88 ± 12.45 ^b^
GPx (mU/mL)	915 ± 58.07 ^a^	449.38 ± 55.79 ^b^	788 ± 180.21 ^a^	784.75 ± 87.95 ^a^
TNF-α (pg/g)	1477.25 ± 149.78 ^a^	2083.13 ± 137.21 ^b^	1490.13 ± 252.35 ^a^	1358.63 ± 139.48 ^a^
IL-1β (pg/g)	938.5 ± 72.06 ^a^	1383.13 ± 170.08 ^b^	1121 ± 151.06 ^a^	1064.25 ± 145.09 ^a^
IL-6 (µg/g)	8.15 ± 1.37 ^a^	14.04 ± 2.82 ^b^	11.16 ± 2.24 ^a,b^	9.43 ± 2.18 ^a^

Different superscript letters (^a^, ^b^, ^c^) in the same row indicate statistically significant differences between groups (*p* < 0.05). Bet3: 3 days of betanin supplementation, Bet15: 15 days of betanin supplementation, MDA: Malondialdehyde, MPO: Myeloperoxidase, SOD: Superoxide dismutase (inhibition rate), GPx: Glutathione peroxidase, TNF-α: Tumor necrosis factor-α, IL-1β: Interleukin-1β, IL-6: interleukin 6.

**Table 2 nutrients-18-00086-t002:** Histological scores of the groups.

Parameters	Control *n* = 8	Colitis *n* = 8	Bet3 + Colitis *n* = 8	Bet15 + Colitis *n* = 8
Mucosal damage (0–3)	0 ± 0 ^a^	3 ± 0 ^b^	2.13 ± 0.64 ^c^	1.5 ± 0.93 ^c^
Cellular infiltration (0–3)	0.25 ± 0.46 ^a^	3 ± 0 ^b^	2.13 ± 0.64 ^c^	1.38 ± 0.74 ^d^

Different superscript letters (^a^, ^b^, ^c^, ^d^) in the same row indicate statistically significant differences between groups (*p* < 0.05). Bet3: 3 days of betanin supplementation, Bet15: 15 days of betanin supplementation.

## Data Availability

The datasets used and/or analyzed during the current study are available from the corresponding author upon reasonable request due to confidentiality and ethical restrictions.

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
