# Peer review of "Protective Effects of Betanin in Acute and Subacute Periods in Experimental Colitis Induced by Trinitrobenzene Sulfonic Acid"

_nutrients, 2025, doi:10.3390/nu18010086_

Round 1

Reviewer 1 Report

Comments and Suggestions for Authors

The manuscript investigates the therapeutic effects of betanin, a natural compound found in red beets, using a TNBS-induced colitis model, with particular emphasis on its ability to modulate inflammation and oxidative stress. The study confirms beneficial effects of betanin supplementation, including notable shift in the levels of inflammatory cytokines and oxidative stress parameters. Evaluating the chemopreventive potential of natural compounds in colitis remains an active and important area of research. Therefore, these findings contribute meaningfully to the ongoing exploration of diet-derived bioactive in gastrointestinal diseases. The study is straightforward and presents interesting and potentially impactful results. However, the manuscript does not include several pieces of essential information needed, especially within the figures and tables.  Moreover, the manuscript would benefit from additional discussion to more clearly articulate the mechanistic implications and strengthen the translational relevance of the work.

Major comments:

  1. A graphical summary of the study design may be helpful to guide readers through the experimental workflow.
  2. It is not clear how the tissues were allocated for biochemical and histological analyses. Were longitudinal halves of the colon used, or were tissue sections selected randomly?
  3. Figure 1 seems to be proximal colon, and others distal. Does TNBS induce colitis in proximal or distal colon?
  4. There are no figure legends included in the manuscript. Clarification is needed on what the arrows and stars represent in the figures.
  5. The tables do not provide any indication of statistically significant differences (i.e. asterisks and/or p-values).
  6. The discussion section largely summarizes previous findings and restates the current results. A deeper integration of the author’s perspective and interpretations would substantially improve the manuscript.

Author Response

Reviewer 1

The manuscript investigates the therapeutic effects of betanin, a natural compound found in red beets, using a TNBS-induced colitis model, with particular emphasis on its ability to modulate inflammation and oxidative stress. The study confirms beneficial effects of betanin supplementation, including notable shift in the levels of inflammatory cytokines and oxidative stress parameters. Evaluating the chemopreventive potential of natural compounds in colitis remains an active and important area of research. Therefore, these findings contribute meaningfully to the ongoing exploration of diet-derived bioactive in gastrointestinal diseases. The study is straightforward and presents interesting and potentially impactful results. However, the manuscript does not include several pieces of essential information needed, especially within the figures and tables.  Moreover, the manuscript would benefit from additional discussion to more clearly articulate the mechanistic implications and strengthen the translational relevance of the work.

Major comments:

Comments 1: A graphical summary of the study design may be helpful to guide readers through the experimental workflow.

Response 1:  Dear Reviewer,

First of all, thank you very much for taking the time to evaluate our manuscript and for your constructive feedback.

A graphic summary has been added to the article in line with your suggestions.

Comments 2: It is not clear how the tissues were allocated for biochemical and histological analyses. Were longitudinal halves of the colon used, or were tissue sections selected randomly?

Response 2: 

Taking your feedback into account, the following sentences have been added to the article.

“…..The cleaned colon tissue was divided lengthwise into two equal parts, and this procedure was performed based on randomly selected sections throughout the tissue. While one half was homogenized for biochemical analysis, the other half was fixed appropriately for histopathological examination. Thus, comparable and representative tissues taken from the same colon were used for both analyses.”

Comments 3: Figure 1 seems to be proximal colon, and others distal. Does TNBS induce colitis in proximal or distal colon?

Response 3: 

Taking your feedback into account, the following paragraph has been added to the article.

“As part of the microscopic histopathological evaluation, the pathologist prepared eight different sections from each tissue sample and took ten microtome samples from each section. After staining these samples, a random sample was selected from each section group for microscopic examination. Since the 10 cm colon segment taken as part of the experimental protocol included both the proximal and distal colon regions, the randomly selected sections contained samples from both anatomical regions. The examinations showed that trinitrobenzene sulfonic acid administration produced histopathological findings associated with ulcerative colitis in a similar manner in both the proximal and distal colon regions. All evaluations were performed using the same longitudinal section analysis method.”

Comments 4: There are no figure legends included in the manuscript. Clarification is nede on what the arrows and stars represent in the figures.

Response 4:  Necessary additions have been made in line with your suggestions.

Comments 5: The tables do not provide any indication of statistically significant differences (i.e. asterisks and/or p-values).

Response 5:  Necessary additions have been made to Tables in line with your suggestions.

Comments 6: The discussion section largely summarizes previous findings and restates the current results. A deeper integration of the author’s perspective and interpretations would substantially improve the manuscript.

Response 6:  Necessary additions have been made in line with your suggestions. 

Sincerely

Reviewer 2 Report

Comments and Suggestions for Authors

This article” Protective and therapeutic effects of betanin in acute and sub-2 acute periods in experimental colitis induced by trinitroben-3 zene sulfonic acid” conducted by Taskiran et al described an experiment that was done with rats to see whether betaine has a protective effect on colitis because it has some properties that reduce inflammation and act as an antioxidant. In this experiment, the rats were put into three different groups: one control group; one group with colitis; and one group receiving three days of betaine supplementation; and one group receiving 15 days of betaine supplementation. Samples were taken from the rats at the end of the experiment and analysed for several biochemical and histopathological parameters, including inflammatory markers (TNF-α and IL-1β), and antioxidant enzyme activities (GPx and SOD). It was found that the supplementation of betaine reduced inflammatory and oxidative markers and improved the damage score based on the histopathologic evaluation at both acute and subacute phases of colitis, suggesting that betaine could be beneficial for the prevention of ulcerative colitis. In particular, the decrease in SOD inhibition rate and MPO level was more pronounced in the subacute phase, indicating that the antioxidant effect of betaine was enhanced over a longer period of supplementation.

The article presents two primary contributions which demonstrate betanin effectiveness in experimental colitis models and establish a new method to study betanin's long-term effects on the body.

  1. First Validation of Betaine Efficacy in a Colitis Model

1-1 Specific Disease Model Validation: The research investigates betaine effects on oxidative stress and antioxidant capacity and inflammatory markers in TNBS-induced experimental colitis models of rats although previous studies have already proven betaine's antioxidant and anti-inflammatory properties.

1-2 The supplement betaine produces strong anti-inflammatory properties which decrease both inflammatory markers and histopathological ratings during the three-day acute period and the fifteen-day subacute period.

1-3 Improved histopathology: Betaine treatment helps protect mucosal tissues from damage and cell infiltration which results in decreased histopathological scores during both acute and subacute stages of colitis.

  1. The study reveals how effects change over time through its analysis of immediate and longer-term betaine administration effects. The research delivers its most important finding through its presentation of complete information about how betaine treatment length affects its resulting effects:

2-1 The study demonstrated that betaine antioxidant properties reached their highest point during the subacute phase which occurred at 15 days.

The research showed that betaine supplementation through Bet15+colitis treatment during the subacute phase brought MPO levels to near-normal rat values but this benefit did not occur during the acute phase.

The subacute supplementation group showed decreased superoxide dismutase (SOD) inhibition as the only result which showed betaine restored SOD activity levels that colitis had decreased.

2-4 The IL-6 levels in the subacute phase were lower than in the colitis group but there was no difference between the two groups during the acute phase.

2-5 The subacute phase group (Bet15+colitis) showed lower cell infiltration fractions than the acute phase group (Bet3+colitis) according to the analysis (p=0.046).

In conclusion, the research makes its discovery of time-dependent mechanisms which show betaine provides optimal benefits for antioxidant and inflammatory markers during “subacute phase supplementation”.

Overall, the core academic expressions, factual descriptions, and statistical results of this academic article are accurate and well-organized, without any obvious or major grammatical or structural errors that might affect comprehension. Most of the "problems" are due to academic writing conventions or formatting flaws.

Some concerns are raised

1.Overstatement: The term "therapeutic" in title is inappropriate. The researchers failed to verify the effectiveness of their treatment approach in their experimental design. The research validated betaine's ability to prevent disease but did not show its effectiveness in treating existing colitis symptoms because researchers started giving betaine only after they caused inflammation.

  1. The study lacked a positive control group to evaluate its findings. The researchers sought to develop new treatment methods for ulcerative colitis but they omitted a positive control group which should have included established anti-inflammatory medications like 5-aminosalicylate and other recognized anti-colitis treatments. The authors used specific references to support their experimental approach and demonstrate that betaine could serve as a therapeutic solution for this disease. The authors did not conduct any experimental group comparisons between the betaine group and the “thyme group” which they used for their anti-colitis-induction mode

3.The study found that betaine treatment failed to produce lasting benefits for particular antioxidant enzymes. The research showed that betaine treatment produced strong antioxidant effects by raising GPx levels during both short-term and extended treatment periods yet the study found no meaningful difference in catalase levels between these time points and colitis rats maintained lower catalase levels than healthy rats. The authors propose that the antioxidant effects of betaine need extended subacute or chronic treatment periods to become visible but this shows that betaine antioxidants do not provide full protection during brief to moderate duration interventions.

  1. Female rats only: The research included 32 female Wistar Albino rats as its only subject group. The study results have limited generalizability because ulcerative colitis pathophysiology might differ between sexes and because female rat hormonal changes could affect their inflammatory and immune system responses. The study results do not indicate how betaine would affect male rats or animals from different models.

  1. Lack of molecular signaling pathway analysis: The research concentrated on studying large-scale indicators which included cytokine activities. However, failed to directly measure betaine's molecular signaling pathways through NF−κB and Nrf2 although it monitored cytokine activities including TNF−α and IL−1β and IL−6 and antioxidant enzyme activities. Research studies have shown that betaine acts as an anti-inflammatory agent through its ability to control NF−κB/PI3K/Akt signaling pathway but this colitis research failed to identify the exact biological processes at play.

Author Response

Reviewer 2

This article” Protective and therapeutic effects of betanin in acute and sub-2 acute periods in experimental colitis induced by trinitroben-3 zene sulfonic acid” conducted by Taskiran et al described an experiment that was done with rats to see whether betaine has a protective effect on colitis because it has some properties that reduce inflammation and act as an antioxidant. In this experiment, the rats were put into three different groups: one control group; one group with colitis; and one group receiving three days of betaine supplementation; and one group receiving 15 days of betaine supplementation. Samples were taken from the rats at the end of the experiment and analysed for several biochemical and histopathological parameters, including inflammatory markers (TNF-α and IL-1β), and antioxidant enzyme activities (GPx and SOD). It was found that the supplementation of betaine reduced inflammatory and oxidative markers and improved the damage score based on the histopathologic evaluation at both acute and subacute phases of colitis, suggesting that betaine could be beneficial for the prevention of ulcerative colitis. In particular, the decrease in SOD inhibition rate and MPO level was more pronounced in the subacute phase, indicating that the antioxidant effect of betaine was enhanced over a longer period of supplementation.

The article presents two primary contributions which demonstrate betanin effectiveness in experimental colitis models and establish a new method to study betanin's long-term effects on the body.

First Validation of Betaine Efficacy in a Colitis Model

1-1 Specific Disease Model Validation: The research investigates betaine effects on oxidative stress and antioxidant capacity and inflammatory markers in TNBS-induced experimental colitis models of rats although previous studies have already proven betaine's antioxidant and anti-inflammatory properties.

1-2 The supplement betaine produces strong anti-inflammatory properties which decrease both inflammatory markers and histopathological ratings during the three-day acute period and the fifteen-day subacute period.

1-3 Improved histopathology: Betaine treatment helps protect mucosal tissues from damage and cell infiltration which results in decreased histopathological scores during both acute and subacute stages of colitis.

The study reveals how effects change over time through its analysis of immediate and longer-term betaine administration effects. The research delivers its most important finding through its presentation of complete information about how betaine treatment length affects its resulting effects:

2-1 The study demonstrated that betaine antioxidant properties reached their highest point during the subacute phase which occurred at 15 days.

The research showed that betaine supplementation through Bet15+colitis treatment during the subacute phase brought MPO levels to near-normal rat values but this benefit did not occur during the acute phase.

The subacute supplementation group showed decreased superoxide dismutase (SOD) inhibition as the only result which showed betaine restored SOD activity levels that colitis had decreased.

2-4 The IL-6 levels in the subacute phase were lower than in the colitis group but there was no difference between the two groups during the acute phase.

2-5 The subacute phase group (Bet15+colitis) showed lower cell infiltration fractions than the acute phase group (Bet3+colitis) according to the analysis (p=0.046).

In conclusion, the research makes its discovery of time-dependent mechanisms which show betaine provides optimal benefits for antioxidant and inflammatory markers during “subacute phase supplementation”.

Overall, the core academic expressions, factual descriptions, and statistical results of this academic article are accurate and well-organized, without any obvious or major grammatical or structural errors that might affect comprehension. Most of the "problems" are due to academic writing conventions or formatting flaws.

Some concerns are raised

Comments 1: Overstatement: The term "therapeutic" in title is inappropriate. The researchers failed to verify the effectiveness of their treatment approach in their experimental design. The research validated betaine's ability to prevent disease but did not show its effectiveness in treating existing colitis symptoms because researchers started giving betaine only after they caused inflammation.

Response 1:  Dear Reviewer,

First of all, thank you very much for taking the time to evaluate our manuscript and for your constructive feedback.

Your feedback has been taken into account, and the term " therapeutic" has been removed from the title.

Comments 2: The study lacked a positive control group to evaluate its findings. The researchers sought to develop new treatment methods for ulcerative colitis but they omitted a positive control group which should have included established anti-inflammatory medications like 5-aminosalicylate and other recognized anti-colitis treatments. The authors used specific references to support their experimental approach and demonstrate that betaine could serve as a therapeutic solution for this disease. The authors did not conduct any experimental group comparisons between the betaine group and the “thyme group” which they used for their anti-colitis-induction mode

 Response 2: In accordance with your criticism, the following paragraph has been added to the manuscript.

“This study has some limitations. First, the absence of a positive control group containing commonly used anti-inflammatory drugs or other known anticolitis agents in the treatment of ulcerative colitis is a significant limitation. Second, the fact that the study was conducted only on female rats limits the generalizability of the findings to both sexes. Finally, the inability to directly measure molecular signaling mechanisms via NF-κB and Nrf2 pathways can be considered another limitation of the study. On the other hand, one of the strengths of this study is that it can serve as a reference for future studies where these limitations are overcome. In addition, being the first validation study of betanin efficacy in a colitis model, the simultaneous examination of acute and subacute periods, and the combined evaluation of oxidant-antioxidant, inflammatory, and histopathological parameters are other strengths of the study.”

Comments 3:The study found that betaine treatment failed to produce lasting benefits for particular antioxidant enzymes. The research showed that betaine treatment produced strong antioxidant effects by raising GPx levels during both short-term and extended treatment periods yet the study found no meaningful difference in catalase levels between these time points and colitis rats maintained lower catalase levels than healthy rats. The authors propose that the antioxidant effects of betaine need extended subacute or chronic treatment periods to become visible but this shows that betaine antioxidants do not provide full protection during brief to moderate duration interventions.

Response 3: Taking your criticism into account, the conclusion section of the summary has been revised as indicated below, and this issue has also been addressed in the discussion section.

“Betanin supplementation demonstrated a significant anti-inflammatory effect by reducing inflammatory parameters and histopathological scores in both periods. Additionally, it exhibited a glutathione-related antioxidant effect by increasing GPx levels in both periods. However, although SOD inhibition rates decreased in the subacute period, no significant change in catalase levels was observed in either period. This indicates that it did not provide complete protection in terms of antioxidant effects in either period.”

Comments 4: Female rats only: The research included 32 female Wistar Albino rats as its only subject group. The study results have limited generalizability because ulcerative colitis pathophysiology might differ between sexes and because female rat hormonal changes could affect their inflammatory and immune system responses. The study results do not indicate how betaine would affect male rats or animals from different models.

 Response 4: In accordance with your criticism, the following paragraph has been added to the manuscript.

“This study has some limitations. First, the absence of a positive control group containing commonly used anti-inflammatory drugs or other known anticolitis agents in the treatment of ulcerative colitis is a significant limitation. Second, the fact that the study was conducted only on female rats limits the generalizability of the findings to both sexes. Finally, the inability to directly measure molecular signaling mechanisms via NF-κB and Nrf2 pathways can be considered another limitation of the study. On the other hand, one of the strengths of this study is that it can serve as a reference for future studies where these limitations are overcome. In addition, being the first validation study of betanin efficacy in a colitis model, the simultaneous examination of acute and subacute periods, and the combined evaluation of oxidant-antioxidant, inflammatory, and histopathological parameters are other strengths of the study.”

Comments 5: Lack of molecular signaling pathway analysis: The research concentrated on studying large-scale indicators which included cytokine activities. However, failed to directly measure betaine's molecular signaling pathways through NF−κB and Nrf2 although it monitored cytokine activities including TNF−α and IL−1β and IL−6 and antioxidant enzyme activities. Research studies have shown that betaine acts as an anti-inflammatory agent through its ability to control NF−κB/PI3K/Akt signaling pathway but this colitis research failed to identify the exact biological processes at play.

Response 5: In accordance with your criticism, the following paragraph has been added to the manuscript.

“This study has some limitations. First, the absence of a positive control group containing commonly used anti-inflammatory drugs or other known anticolitis agents in the treatment of ulcerative colitis is a significant limitation. Second, the fact that the study was conducted only on female rats limits the generalizability of the findings to both sexes. Finally, the inability to directly measure molecular signaling mechanisms via NF-κB and Nrf2 pathways can be considered another limitation of the study. On the other hand, one of the strengths of this study is that it can serve as a reference for future studies where these limitations are overcome. In addition, being the first validation study of betanin efficacy in a colitis model, the simultaneous examination of acute and subacute periods, and the combined evaluation of oxidant-antioxidant, inflammatory, and histopathological parameters are other strengths of the study.”

Sincerely

Round 2

Reviewer 1 Report

Comments and Suggestions for Authors

The authors have addressed the majority of the reviewers' comments. However, the discussion and conclusion are still somewhat lengthy and include repetitive descriptions of the methods and results. Shortening these sections may improve the overall quality of the manuscript. 

Author Response

Dear Editor,

First, we would like to thank you for sending our manuscript for external peer review. We would like to thank the reviewers also for their constructive criticism. The reviewers’ comments and our responses are listed below.

Sincerely yours

Reviewer 1

Comments 1: The authors have addressed the majority of the reviewers' comments. However, the discussion and conclusion are still somewhat lengthy and include repetitive descriptions of the methods and results. Shortening these sections may improve the overall quality of the manuscript. 

Response 1:  Dear Reviewer,

First of all, thank you very much for taking the time to evaluate our manuscript and for your constructive feedback.

As you suggested, the discussion and conclusion sections have been shortened.

Sincerely
